# Ultrasound-Guided Needle Aspiration of Peritonsillar Abscesses: Utility of Transoral Pharyngeal Ultrasonography

**DOI:** 10.3390/diagnostics9040141

**Published:** 2019-10-05

**Authors:** Yuta Hagiwara, Yoshimitsu Saito, Hana Ogura, Yuichiro Yaguchi, Takahiro Shimizu, Yasuhiro Hasegawa

**Affiliations:** 1Department of Neurology, St. Marianna University School of Medicine, Kawasaki 216-8511, Japan; ogura-h@marianna-u.ac.jp (H.O.); Shimi-taka@marianna-u.ac.jp (T.S.); hasegawa-neuro1@marianna-u.ac.jp (Y.H.); 2Department of Otorhinolaryngology, St. Marianna University School of Medicine, Kawasaki 216-8511, Japan; y2saitou@marianna-u.ac.jp (Y.S.); yaguchi@marianna-u.ac.jp (Y.Y.)

**Keywords:** transoral ultrasonography, peritonsillar abscess, puncture drainage

## Abstract

A peritonsillar abscess is a common deep infection that is usually related to acute tonsillitis. Needle aspiration is often performed for diagnosis and treatment, but several complications, including puncture of the carotid artery, may occur, even when performed by properly trained physicians. The utility of transoral pharyngeal ultrasonography (TOPU) equipped with a biopsy adaptor for safe and full aspiration is presented. A 19-year-old man was admitted to our hospital because of a peritonsillar abscess. TOPU showed the abscess and a branch of the carotid artery, and an otolaryngologist performed puncture through the biopsy adaptor with the aid of the ultrasound image. Needle aspiration was accomplished by avoiding arterial puncture and monitoring the shrinkage of the abscess. TOPU-guided needle aspiration is useful in the safe drainage of peritonsillar abscesses.

## 1. Introduction

Transoral ultrasonography is a well-known technique for examining the carotid artery [1,2,3,4], known as “transoral carotid ultrasonography” (TOCU). Transoral ultrasonography is also useful in the field of otolaryngology, in addition to angiology [5,6,7]. In transoral ultrasonography, a transvaginal probe is inserted into the mouth and then placed on the wall of the pharynx. For otolaryngology, we have termed this technique “transoral pharyngeal ultrasonography” (TOPU) [8] to distinguish it from TOCU. A peritonsillar abscess is a common infectious disease, with an incidence of 30/100,000 people per year in the United States [9]. Surgical management is indicated in a case of complicated peritonsillar abscess, consisting of early treatment with antibiotics and drainage of the pus [10]. Otolaryngologists usually perform needle aspiration blindly, but several insertions of the needle at different points might be needed. Needle aspiration may not always be successful, especially when the volume of pus could be small. TOPU can show the locations of the abscess and arteries, and it might improve the safety of drainage. A case of peritonsillar abscess is presented, illustrating the utility of TOPU-guided needle aspiration.

## 2. Case Report

A 19-year-old man was admitted to our hospital because of pharyngeal pain with fever. A peritonsillar abscess was detected on contrast-enhanced computed tomography on the left side of the palatine tonsil (Figure 1). An otolaryngologist tried to perform needle aspiration of the abscess blindly, but it was unsuccessful. Otolaryngologists then decided to perform needle aspiration of the abscess and then consulted us for evaluation by transoral ultrasonography. TOPU was performed while using a biopsy adaptor (UAGV-024A, Canon Medical Co., Tochigi, Japan) and a custom-ordered 21-gauge/25-cm needle (Figure 2a,b), similar to a percutaneous transhepatic gallbladder aspiration needle. An Ultrasound Aplio500 (Canon Medical Co.) and a 6-MHz transvaginal probe were used for ultrasound examination. A long needle was prepared for drainage since the length of the probe was 20 cm. The patient was placed in the sitting position during TOPU. Under local anesthesia of the pharynx, the probe was inserted into the mouth while avoiding the tongue and pressed softly against the anterior palatine arch (Figure 3). TOPU showed the locations of the abscess and the carotid arteries (Figure 4a). The needle was clearly visualized on the images, and the otolaryngologist inserted the needle into the center of the abscess while avoiding arterial puncture (Figure 4b,c, and Appendix A). Another otolaryngologist aspirated the pus by the syringe that was connected to the needle with an extension tube until the hypoechoic area disappeared on the image. In total, 2.4 mL of pus was aspirated, and the otolaryngologist made an incision at the puncture point for further drainage. The patient’s severe throat pain and speech disturbance improved shortly after the aspiration, and the patient had a good clinical course without recurrence.

## 3. Discussion

Previously, we reported TOPU as a tool for determining the direction and depth of puncture for peritonsillar abscesses [8]. In this patient, TOPU equipped with a biopsy adaptor specially designed for biopsy of uterine appendages was used. This system has the advantage of having a large field of view and depicting a guided line system for the needle in the images. In blind puncture of the pharynx, mistaken puncture of the carotid arteries is one of the serious complications. It has also been reported that blind puncture often does not result in insertion into the space with the pus; therefore, blind puncture was reported to have a false-negative rate for the diagnosis of peritonsillar abscess of 10–24% [11,12]. The space, including the pus in a peritonsillar abscess, is demonstrated as a low echoic space on transoral ultrasound; therefore, the operator can precisely puncture the pus. The abscess, needle, and arteries should be simultaneously demonstrated on one image during ultrasonography to achieve safe and accurate drainage. TOPU was developed based on TOCU, which is useful for evaluating the carotid arteries; therefore, TOPU can show the abscess and needle, in addition to the carotid artery and its branches, at the same time.

Blaivas et al. reported ultrasound-guided drainage of peritonsillar abscesses while using an endocavity probe [6,7]. The Blaivas technique clearly depicts peritonsillar abscesses by ultrasound. The advantage of TOPU over the Blaivas technique is that the operator can insert the needle parallel to the probe through the biopsy attachment in TOPU, which might be useful for inexperienced operators. The utility of transoral ultrasonography for the drainage of peritonsillar abscesses using a pencil-shaped Burr-Hole transducer and a 10-MHz small linear probe (hockey stick probe) has been demonstrated in other reports [13,14]. The attachment of a needle guide similar to the present TOPU method was used in the report of the pencil-shaped Burr-Hole transducer method [13]. These methods might have difficulty visualizing deep structures due to the narrow field of view in the image. Although the low-frequency endocavity (transvaginal) probe decreases axial image resolution as compared to a high-frequency probe, such as a hockey stick probe, the low-frequency probe is suitable for the observation of deep structures, such as the carotid arteries, providing a large field of view in the image.

The hockey stick probe and the Burr-Hole probe may be suitable for a surgical procedure due to the small probe size. As patients with a peritonsillar abscess may have trismus, a smaller probe suitable for needle aspiration is appropriate in such cases.

TOPU-guided needle aspiration appears to be useful for safe drainage of peritonsillar abscesses.

## Figures and Tables

**Figure 1 diagnostics-09-00141-f001:**
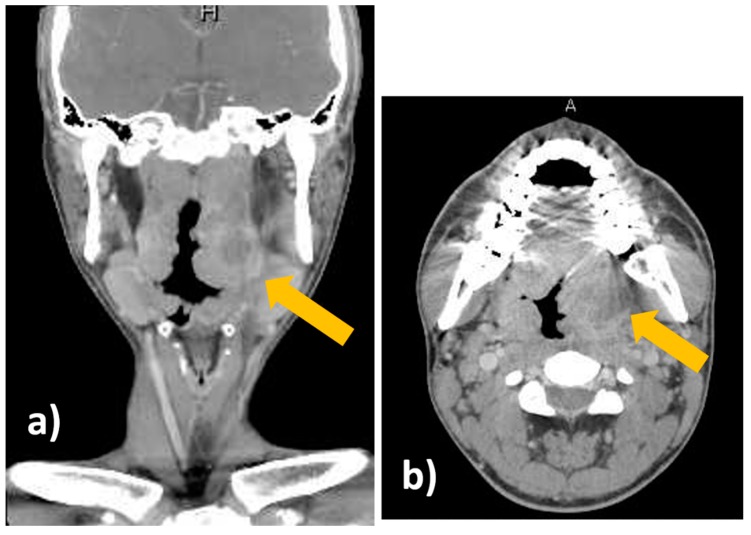
Contrast-enhanced computed tomography. (**a**) Coronal view. (**b**) Axial view. The peritonsillar abscess is indicated by the yellow arrows.

**Figure 2 diagnostics-09-00141-f002:**
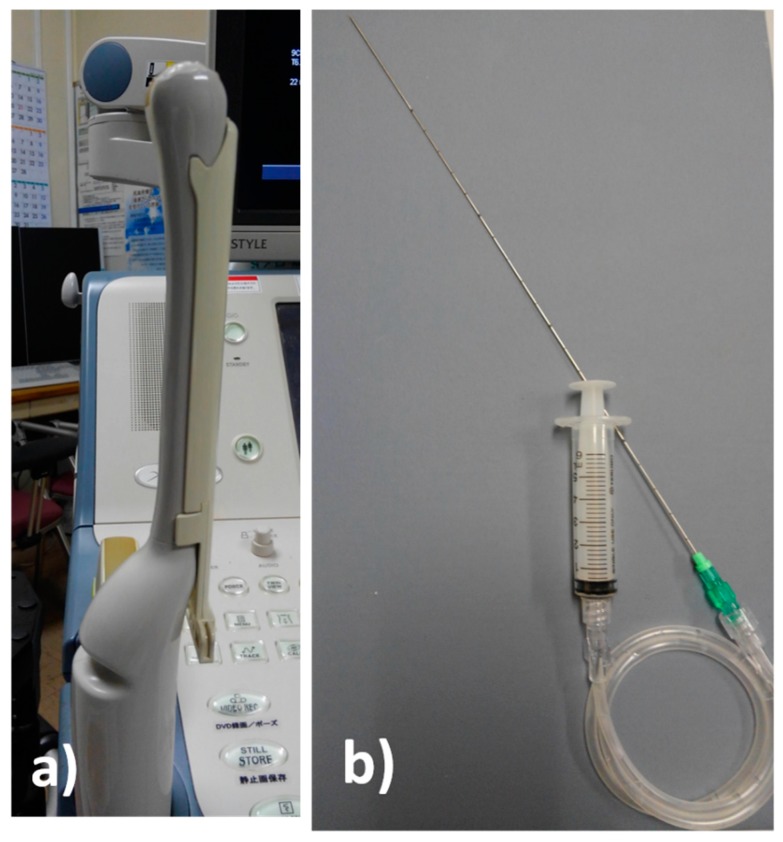
The preparation for transoral pharyngeal ultrasonography (TOPU). (**a**) The 6-MHz transvaginal probe with biopsy adaptor (UAGV-024A, Canon Medical Co., Tochigi, Japan). (**b**) The custom-ordered 21-gauge, 25-cm needle with syringe and extension tube.

**Figure 3 diagnostics-09-00141-f003:**
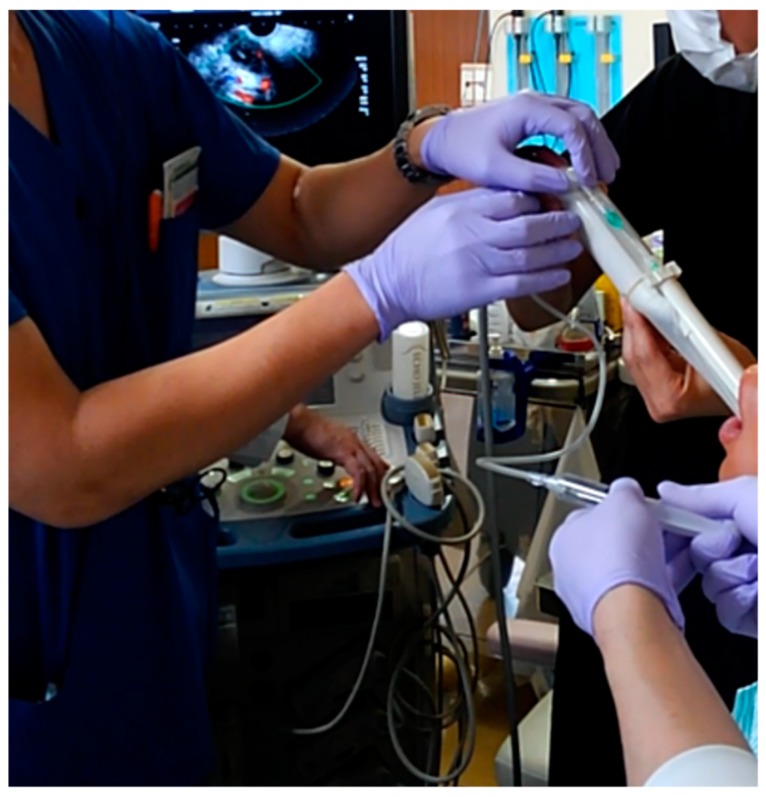
Puncture drainage by transoral pharyngeal ultrasonography (TOPU). Three doctors are performing the puncture drainage with TOPU. The doctor in the right upper side of the image is a neurologist, and he is operating the probe. The doctor on the left side of the image is an otolaryngologist, and he is inserting the needle into the pharynx under ultrasound imaging guidance. Another otolaryngologist is on the right lower side of the image, and he is aspirating pus with the 5-cc syringe.

**Figure 4 diagnostics-09-00141-f004:**
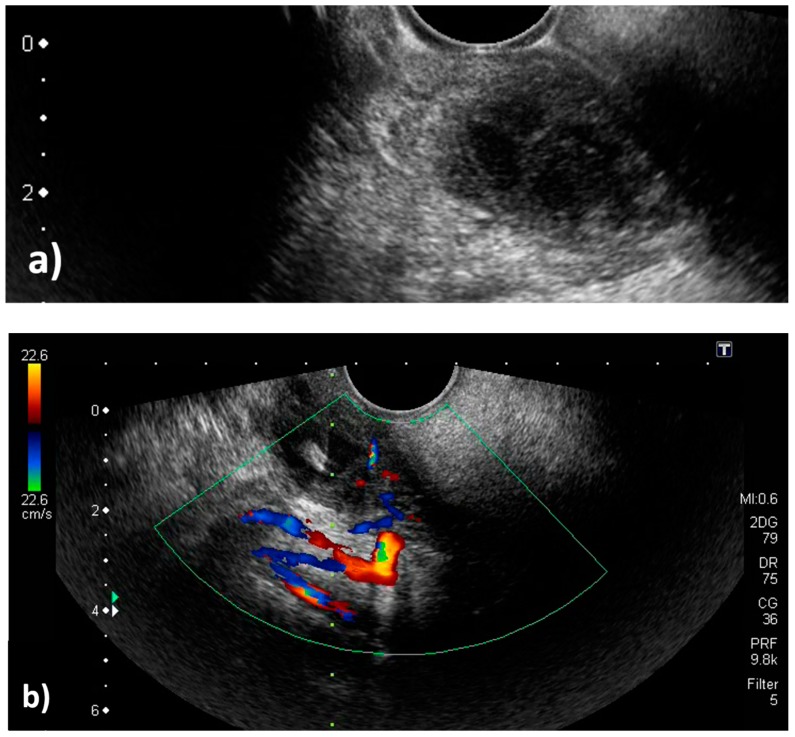
Transoral pharyngeal ultrasonography (TOPU). (**a**) TOPU with B-mode. (**b**) TOPU with color Doppler. The left side of the image is the head side, and the right side is the foot side. TOPU shows the internal carotid artery (ICA) under the abscess at a depth of 30–40 mm and the external carotid artery (ECA) just under the abscess at a depth of 20–30 mm. The branches of the ECA surround the head side of the abscess. The tip of the needle is inserted at the center of the abscess in the hypoechoic space. The vertical green dotted line indicates the direction in which the needle advances; this guiding line system is incorporated in the ultrasound machine (Aplio500). (**c**) Schematic image. The periphery of the abscess is indicated by the red circle. It is easy to differentiate between the ICA and ECA because the branch arteries are visualized along with the ECA.

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
