# Peer review of "Ultrasound-Guided Needle Aspiration of Peritonsillar Abscesses: Utility of Transoral Pharyngeal Ultrasonography"

_diagnostics, 2019, doi:10.3390/diagnostics9040141_

Round 1

Reviewer 1 Report

This case report describes a case where the authors use a new technique for ultrasound-guided needle aspiration of peritonsillar abscesses using an endocavity transducer with biopsy adaptor. Many previous studies have described the use of endocavity transducer to improve the diagnostic work-up of peritonsillar abscesses, but only very few studies describe the ultrasound-guided needle aspiration. With this technique, needle aspiration of peritonsillar can safely be performed, and vascular structures identified. The use of an endocavity transducer may be more available on standard ultrasound equipment at the hospital than other more specialized probes like the hockey stick, and burr-hole transducer described earlier. However, there are several limitations of this technique which need to be elaborated in the discussion.

General comments:

Did you only perform needle aspiration as a treatment? What about afterward incision with a knife or a pean for drainage?

Did the patient return with peritonsillar abscess, or was he successfully treated with one needle aspiration?

Six authors to a single case report. Can you please describe the role of the different authors?

Other comments:

Line 11: I would not call the needle aspiration for the gold standard for both diagnostics and treatment of peritonsillar abscesses.

Line 32: “this method is often dangerous due to mistaken puncture of arteries.”
I do not agree! Needle aspirations are performed very frequently and have a very low complication rate. Puncture of arteries is extremely rare. However, peritonsillar Abscesses are challenging to diagnose on the clinical exam only, why many needle aspirations may be unsuccessful because of the lack of a peritonsillar abscess.

Line 39 Why did you need ultrasound when you had a contrast-enhanced computed tomography? Was it difficult to reach the abscess cavity blindly?

Line 40: Custom made needle? Please explain further. Do you think you would be able to aspirate a peritonsillar abscess with more thick pus with such a long needle?

Line 47: how many ml Pus were aspirated? Do you have any measures of the abscess cavity on ultrasound / CT? Do you have an ultrasound video clip of the aspiration of the peritonsillar abscess?

Line 94: What about the many patients with trismus? Many of these patients have difficulty with mouth opening, making TOPU difficult with an endocavity transducer?

Line 97: It is not my experience that it is difficult to see the vascular structures with a hockey stick or burr-hole transducer. You need to use low frequency to visualize deep structures like the carotid artery. Instead, the low frequency of the endocavity transducer (6mhz) decreases axial image resolution of the imaging compared to high-frequency ultrasound transducer like the hockey stick. Please elaborate on what this means to the diagnostic work-up in the discussion.

Figure 4. Please describe how you differentiate the ICA and ECA on ultrasound. It is not evident to me.

Reviewer 2 Report

The authors have reported the use of transoral pharyngeal ultrasonography for drainage of peritonsillar abscess in a 19-year old patient. In contrast to TOCU which is useful in evaluating carotid arteries, the authors state that TOPU, in addition to showing the carotid artery, also shows the abscess and the aspiration needle at the same time on a sonograph. The author have previously published a case report in which they determine the direction and depth of puncture based on the information of TOPU imaging.

Following are my comments:

(1) Figure legend 3 is very confusing regarding which physician is being referred to. Please rephrase it. 

(2) Some sentences appear very similar to the ones in their previous article in Clinical Case Reports. Please modify.

(3) Have the authors tested the usefulness of TOPU in this patient only or it has been used in more patients?

(4) There are some odd sentences throughout the manuscript, and should be checked for grammatical corrections.
